# Modelling the impact of JNJ-1802, a first-in-class dengue inhibitor blocking the NS3-NS4B interaction, on in-vitro DENV-2 dynamics

Clare P. McCormack[1]*, Olivia Goethals[2], Nele Goeyvaerts[3], Xavier D. Woot de Trixhe[3], Peggy Geluykens[3,4], Doortje Borrenberghs[2], Neil M. Ferguson[1], Oliver Ackaert[3‡]*, Ilaria Dorigatti [1‡]

**1** MRC Centre for Global Infectious Disease Analysis, School of Public Health, Imperial College London, London, United Kingdom, **2** Janssen Global Public Health, Janssen Pharmaceutica NV, Beerse, Belgium, **3** Janssen Research & Development, Janssen Pharmaceutica NV, Beerse, Belgium, **4** Discovery, Charles River Beerse, Beerse, Belgium

‡ These authors are joint senior authors on this work.
* c.mccormack14@imperial.ac.uk (CPM); oackaert@ITS.JNJ.com (OA)

**Data Availability Statement:** All data and code supporting the findings of this study are publicly available at https://github.com/claremccormack/

## Abstract

Dengue virus (DENV) is a public health challenge across the tropics and subtropics. Currently, there is no licensed prophylactic or antiviral treatment for dengue. The novel DENV inhibitor JNJ-1802 can significantly reduce viral load in mice and non-human primates. Here, using a mechanistic viral kinetic model calibrated against viral RNA data from experimental *in-vitro* infection studies, we assess the *in-vitro* inhibitory effect of JNJ-1802 by characterising infection dynamics of two DENV-2 strains in the absence and presence of different JNJ-1802 concentrations. Viral RNA suppression to below the limit of detection was achieved at concentrations of >1.6 nM, with a median concentration exhibiting 50% of maximal inhibitory effect ($IC_{50}$) of $1.23 \times 10^{-02}$ nM and $1.28 \times 10^{-02}$ nM for the DENV-2/RL and DENV-2/16681 strains, respectively. This work provides important insight into the *in-vitro* inhibitory effect of JNJ-1802 and presents a first step towards a modelling framework to support characterization of viral kinetics and drug effect across different host systems.

## Author summary

Dengue poses a major challenge to global public health, with almost half of the world's population at risk of infection each year. However, currently, no licensed specific prophylactic or antiviral treatment for dengue exists. JNJ-1802, a novel dengue inhibitor under development by Janssen Pharmaceutica, is one of the most advanced candidate dengue antivirals and has recently been shown to significantly reduce viral load in pre-clinical animal studies. Here we use mathematical models, fitted to viral RNA data obtained from experimental infection studies in Vero cells, to quantify the *in-vitro* inhibitory effect of JNJ-1802 on the viral kinetics of two strains of dengue serotype 2. For both strains, we find the effect of JNJ-1802 on viral replication is strongly concentration-dependent, with viral suppression to below the limit of detection achieved at concentrations of 1.6 nM and

DENV_In_Vitro. Source data are also provided with this paper (S1_Data).

**Funding:** CMC, NMF and ID acknowledge funding from the MRC Centre for Global Infectious Disease Analysis (reference MR/X020258/1), funded by the UK Medical Research Council (MRC). This UK funded award is carried out in the frame of the Global Health EDCTP3 Joint Undertaking. ID acknowledges funding by the Wellcome Trust (grant number 213494/Z/18/Z). This research was partially funded by Janssen Pharmaceutica and VLAIO (DENVTrans project HBC.2019.2906). For the purpose of open access, the authors have applied a CC BY public copyright licence to any Author Accepted Manuscript version arising from this submission. Janssen Pharmaceutica performed the experiments and data collection, assisted with preparation of this manuscript. Other funders had no role in study design, data collection and analysis, decision to publish, or preparation of the manuscript.

**Competing interests:** I have read the journal's policy and the authors of this manuscript have the following competing interests: OG, NG, XWT, DB, OA are employees of Janssen Pharmaceutica NV and potential shareholders of Johnson & Johnson. ID received a research grant funded by Janssen to conduct this study.

above. This work is an important first step in quantifying the potential of JNJ-1802 as a dengue antiviral, and demonstrates how interdisciplinary research can help inform drug development and help inform dosing regimens.

## Introduction

Dengue is an acute, systemic mosquito-borne infection caused by one of four antigenically distinct but closely related dengue virus serotypes (DENV-1-4) [1]. As the world's most prevalent arboviral infection in humans, it poses a major public health challenge in tropical and subtropical regions worldwide [2–4]. The clinical spectrum of dengue disease is wide, with manifestations ranging from asymptomatic or mild illness to severe, life-threatening conditions including haemorrhagic fever and shock syndrome [1,5]. Recent estimates suggest approximately 51 million symptomatic dengue infections occur globally each year [4], with outbreaks of dengue increasing in both size and duration in recent years [6]. Currently, there is no specific prophylactic or antiviral treatment for dengue [7,8], and the only available option is supportive care. Furthermore, while the QDENGA vaccine has been recently approved by the European Medicines Agency (EMA), in Indonesia [9], Brazil [10], Argentina [11], Thailand [12] and in the UK [13], the impact of vaccination at the population level still needs to be evaluated. Vector control also remains challenging given concerns over the feasibility of implementing novel control strategies such as the release of *Wolbachia*-infected mosquitoes in resource-constrained settings [3,14–16]. Hence, specific antiviral treatment is urgently needed for the control of dengue disease both at the individual and population level.

A strong association exists between disease severity and viral load for dengue [17–19], with patients suffering from dengue shock syndrome having viral titres up to 10- to 100-fold higher than those with mild disease [18]. Therefore, it is hypothesised that an antiviral which reduces viral load will consequently reduce the severity and duration of symptoms and the risk of progression to severe disease [20–22]. In recent decades, the search for an effective dengue antiviral has followed two main pathways–repurposing drugs, licensed to treat other infections and conditions, for dengue treatment, and developing a novel dengue-specific antiviral [8,22–26]. Clinical trials investigating the use of existing drugs for dengue treatment have thus far been unsuccessful in finding a suitable candidate for repurposing [7,8,22,23], with drugs including balapiravir [27] and celgosivir [28] (hepatitis C), chloroquine [29] (malaria), and ivermectin [26,30] (parasitic infections) all proving clinically ineffective against dengue in humans. Strategies to develop a novel dengue-specific antiviral have involved either targeting cellular pathways (host-directed antivirals or HDAs) or structural and non-structural viral proteins essential to the virus life cycle (direct-acting antivirals or DAAs) [7,21,22,24,25,31,32]. While several potential HDAs have been identified, to date, few have been evaluated in clinical trials [7]. Among the non-structural proteins facilitating viral replication [33–35], antiviral research has primarily focused on NS5 and NS3 [7,24] and, in recent years, NS4B [7,24,36–42].

JNJ-1802 is a novel small DAA molecule under development by Janssen Pharmaceutica, from the same chemical space as JNJ-A07 [42], which inhibits DENV replication by blocking the interaction between NS3 and NS4B within the viral replication complex [43]. While clinical studies assessing the efficacy of JNJ-1802 are currently underway [43], the potential of JNJ-1802 as a dengue antiviral has recently been demonstrated, with *in-vitro* and pre-clinical animal studies showing JNJ-1802 to be a highly specific, potent DENV inhibitor in multiple host systems, with a favourable toxicity profile and high barrier to resistance [43].

The aim of this study was to estimate the *in-vitro* prophylactic inhibitory effect profile of JNJ-1802 from infection experiments conducted in Vero cells using mechanistic viral kinetics models. The use of mechanistic models based on ordinary differential equations for reconstructing the within-host dynamics of viral infections in humans [44–51], including dengue [46,52–55], have been demonstrated to be a powerful tool to help build our understanding of viral kinetics, the host immune response, as well as antiviral drug effects[56]. One of the primary advantages of adopting a mechanistic modelling approach when assessing the impact of JNJ-1802 is that it allows us to make full use of longitudinal viral kinetic data, and this in turn allows us to estimate the antiviral concentration required for viral suppression in Vero cells. In this paper we present data on the viral kinetics of two DENV-2 strains from *in-vitro* infection experiments in Vero cells, in the absence and presence of JNJ-1802, and use mathematical models to characterise the *in-vitro* kinetics of dengue virus, quantify the *in-vitro* inhibitory effect of JNJ-1802, and assess its mechanism of action. This work highlights how interdisciplinary research and mathematical modelling applied to pre-clinical data can help identify promising drug candidates from the early phases of drug development and help inform dosing regimens.

## Results

To assess the inhibitory effect of JNJ-1802 on the *in-vitro* viral kinetics of dengue infection, experiments in Vero cells were carried out by infecting the cells on day 0 with either the DENV-2/16681 or DENV-2/Rega Lab (RL) strain. Intracellular and extracellular viral RNA levels in the absence and presence of JNJ-1802 at different concentrations were measured daily for 10 days, and full details of the experiments conducted and data collected are provided in the Methods section below. Experiments were conducted with and without refreshing the medium of individual wells to test the sensitivity of the observed viral load profiles to underlying experimental conditions. Except for a limited number of measurements, which were inconsistent with the overall pattern in viral growth observed both at the intracellular and extracellular level (Fig 1), we observed a relatively small degree of variation between individual wells for both the intracellular and extracellular RNA data for the DENV-2/16681 and DENV-2/RL strains (Fig A in S1 Text). In addition, similar parameter estimates were obtained for experiments using the same strains where the well medium was/was not refreshed on day 4 post infection (p.i.), thereby indicating that the kinetics observed were not owing to changes in underlying experimental conditions.

For both strains, we quantified the impact of JNJ-1802 on infection in Vero cells by fitting a mechanistic viral kinetics model of the dynamics of DENV infection *in-vitro* simultaneously to the intracellular and extracellular RNA data observed from the experimental infection studies. The model describes the infection process of Vero cells and subsequent secretion of viral RNA, and assumes JNJ-1802 acts on the transition process of infected to infectious (virion producing) cells. Full details of the model developed, parameter values, and model fitting approach are provided in the Methods section below.

Below we present the results obtained in the experiments conducted without medium refresh, unless specified otherwise. The results obtained for the experiments where the medium of the cells was refreshed on day 4 are provided in Table A and Figs B-D in S1 Text.

### Viral kinetics of DENV-2 Infection

In control conditions without JNJ-1802 present, both the DENV-2/16681 and DENV-2/RL strains showed similar temporal trends in viral replication, with intracellular viral RNA measurements above the limit of detection first observed two days following inoculation, thereby

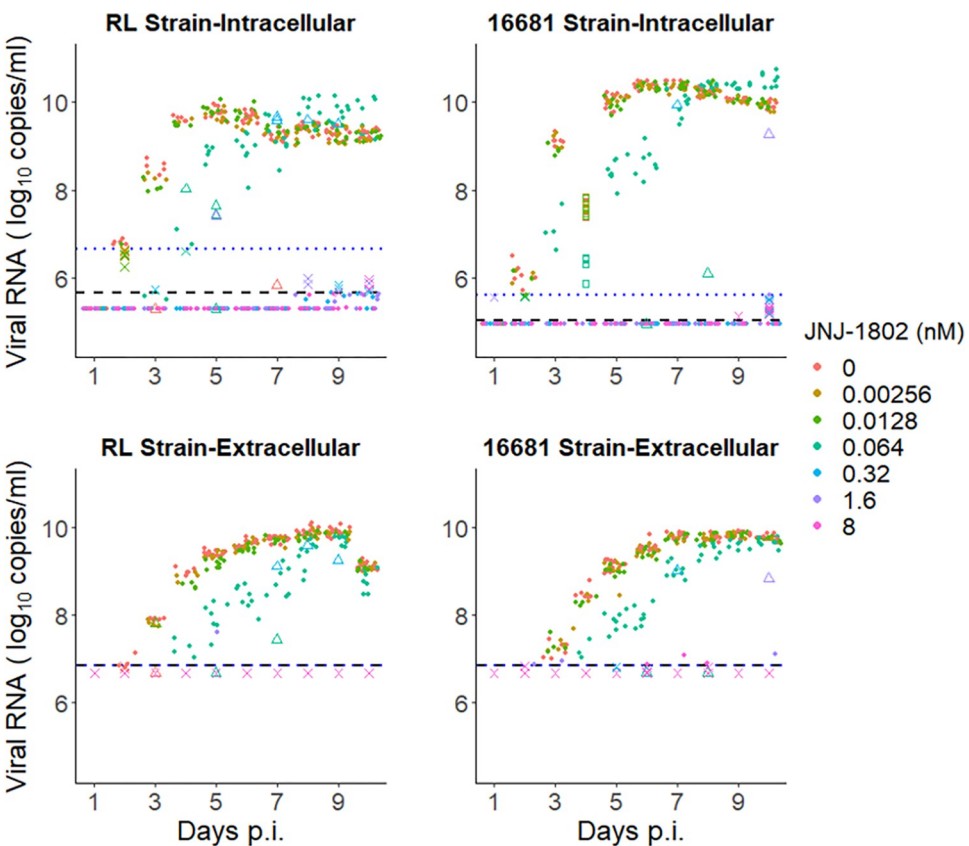

**Fig 1. Summary of viral load data from the *in-vitro* infection experiments in Vero cells.** Summary of all data collected, disaggregated by the infecting DENV-2 strain (RL, 16681), JNJ-1802 concentration and type of viral RNA (intra- or extracellular). Measurements with $> 1 \log_{10}$ difference from the corresponding median (triangles) were not used for model fitting. We also excluded the intracellular data collected on day 4 for experiments using the DENV-2/ 16681 dengue strain and antiviral concentrations of 0 nM, $2.56 \times 10^{-03}$ nM, $1.28 \times 10^{-02}$ nM, and $6.40 \times 10^{-02}$ nM, as these were inconsistent with the corresponding extracellular data (18 measurements in total, squares in topright subfigure). The limit of detection (LOD) and the limit of quantification (LOQ) are indicated by horizontal black and blue lines, respectively. For extracellular data, these limits were identical. Crosses represent measurements below the limit of quantification which were assumed to be left censored in a sensitivity analysis.

typically peaking by day five to seven and decreasing slowly thereafter (Figs 1–3). Extracellular viral RNA profiles followed a similar overall pattern, with a lag of approximately one day compared to intracellular viral RNA (Figs 1–3). We estimated depletion of target cells by day 4 p.i. (Figs E and F in S1 Text).

Figs 2 and 3 show the model fit to the viral RNA measurements over time observed for the DENV-2/16681 and DENV-2/RL strain respectively, using the developed viral kinetic model with an antiviral effect on the transition of infected cells to infectious (i.e., virion producing) cells (more details can be found in the Methods section). Table 1 shows the corresponding posterior estimates of the model parameters. The viral kinetics model generally captures well the observed intracellular and extracellular viral RNA profiles for both the DENV-2/16681 and DENV-2/RL strains in the absence and presence of JNJ-1802 (Figs 2, 3, and G in S1 Text). However, for some concentrations, viral RNA levels were over- or under-estimated by the model at several timepoints (Figs 2 and 3). For example, for a JNJ-1802 concentration of 0.064nM the model under-estimates viral RNA levels on day 3 p.i. for the 16681 strain (Fig 3) and for a JNJ-1802 concentration of 0.32 nM, the model predicts intracellular RNA above the

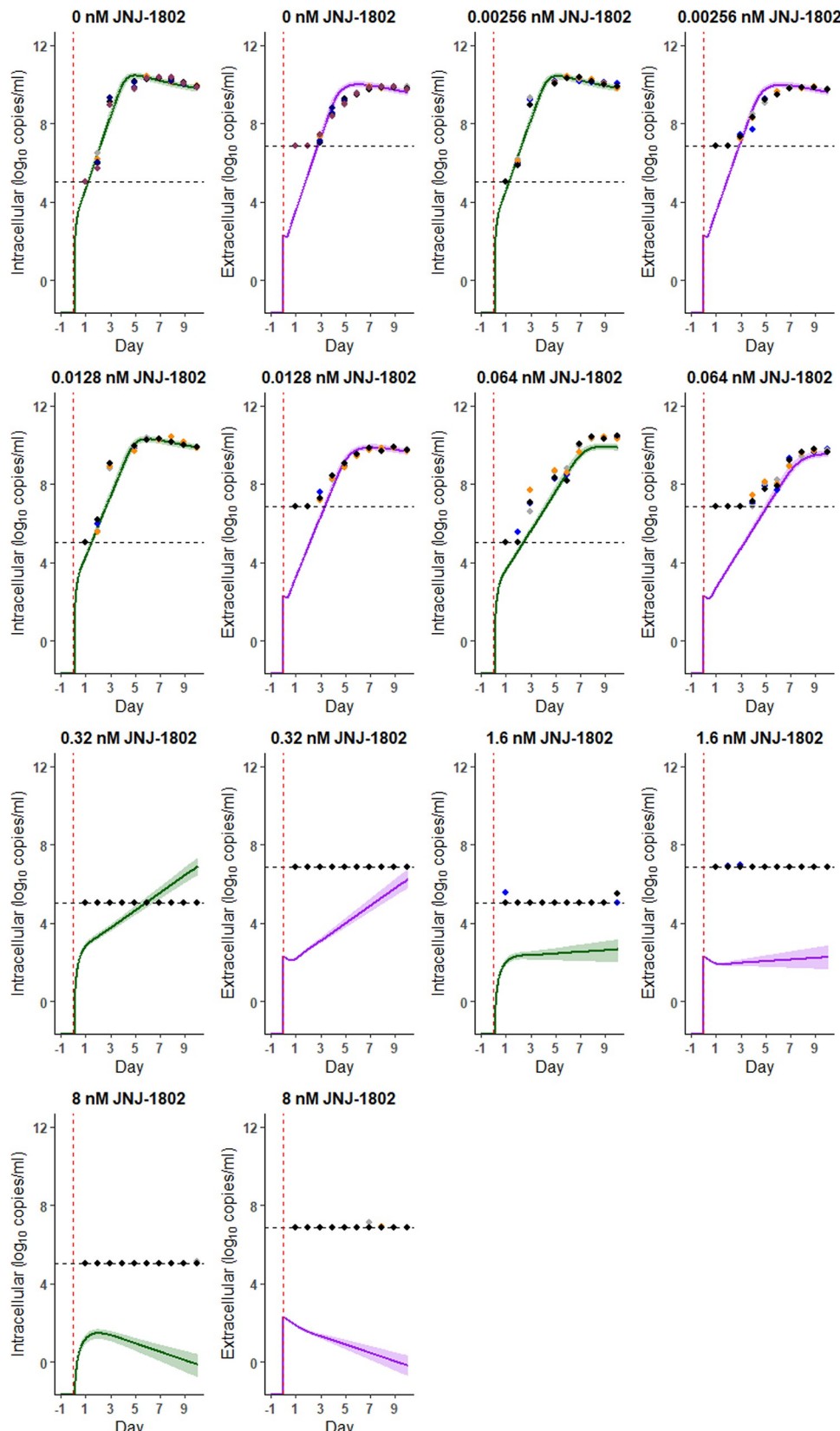

**Fig 2. Model Fits (DENV-2/16681 strain).** Model fits for the measurements observed using the DENV-2/16681 strain with no medium refresh and antiviral concentrations of 0 nM, $2.56\times10^{-03}$ nM, $1.28\times10^{-02}$ nM, $6.40\times10^{-02}$ nM, 0.32 nM, 1.6 nM, and 8 nM JNJ-1802. Coloured points represent the data from each well (6 wells for 0 nM, 4 wells for concentrations >0 nM), the vertical red line indicates the time the viral inoculum was added to each well and the horizontal dashed black line indicates the limit of detection (LOD). The modelled dynamics of the intracellular RNA virus are in green and those of the extracellular RNA virus are in purple; solid lines represent the median and the shading represents the 95% CrI. Here, the measurements below the limit of quantification (LOQ) (crosses in Fig 1) were included during model fitting. Measurements below LOD were left-censored at the LOD during model fitting and we plot these measurements at the LOD for visual display. We assumed that the antiviral directly inhibits the transition process of infected cells to infectious (virion producing) cells i.e., acts on τ.

limit of detection from seven days p.i. onwards, when the majority of corresponding intracellular RNA measurements remained below this limit for the duration of the experiments (Figs 2, 3, B, and C in S1 Text). Although the temporal trends in viral replication were similar for the DENV-2/16681 and DENV-2/RL strains, we estimated an approximately two- to three-fold difference between strains in the rate of intracellular viral RNA production, with a substantially higher rate for the DENV-2/16681 strain (median: $1.36\times10^7$ per infectious cell per day) compared with the DENV-2/RL strain (median: $3.17\times10^6$ per infectious cell per day) (Table 1). However, in the same experiment, we estimated the infection rate of target cells was slightly higher for the DENV-2/RL strain (median: $3.18\times10^{-08}$ per virion per day) compared with the DENV-2/16681 strain (median: $1.32\times10^{-08}$ per virion per day). Our results also suggest that a larger proportion of intracellular RNA was subsequently released as extracellular RNA for the DENV-2/RL strain (medians: 59% (DENV-2/RL), 23% (DENV-2/16681) (Table 1). Estimates of the proportion of intracellular RNA subsequently released as extracellular RNA were negatively correlated with both the target cell infection rate and the rate of intracellular viral RNA production corresponding to the balance between viral kinetics at the intracellular level (cell infection, viral RNA production) and those at the extracellular level (viral RNA release) required to match the dynamics observed. Overall, our estimates suggest that the DENV-2/RL strain had a higher viral fitness, as measured by the basic reproduction number, than the DENV-2/16681 strain, with an $R_0$ of 277.01 (95% CrI: 231.78, 332.66) for the DENV-2/RL strain compared to 191.10 (95% CrI: 161.84, 232.76) for the DENV-2/16681 strain (Table 1).

## JNJ-1802 Inhibitory Effect

For both strains, the model estimated suppression of the virus to below the limit of detection over the course of the experiment, at both the intracellular and extracellular level, for a concentration of 1.6 nM or greater (Figs 2 and 3). A strong concentration-dependent inhibitory effect of JNJ-1802 on viral replication was estimated across all experiments, approximating 100% inhibition at concentrations of 1.6 nM or greater (Fig 4). Estimates of the $IC_{50}$, Hill coefficient and the corresponding concentration-effect profiles varied slightly between strains (Table 1 and Fig 4). We estimated a similar $IC_{50}$ for both strains (median: $1.23\times10^{-02}$ nM (DENV-2/RL), $1.28\times10^{-02}$ (DENV-2/16681)) and a slightly higher median Hill coefficient for the DENV-2/RL strain (median: 1.28) compared with the DENV-2/16681 strain (median: 1.17) (Table 1). However, the 95% CrI intervals for estimates of the Hill coefficient were overlapping (95% CrI: (1.14, 1.42) (DENV-2/RL), (1.07, 1.31) (DENV-2/16681)). The effective reproduction number was estimated to reach 1 at a JNJ-1802 concentration of approximately 2.02 nM (95% CrI: 1.52 nM, 2.87 nM) and 2.43 nM (95% CrI: 1.77 nM,3.43 nM) for the DENV-2/RL and DENV-2/16681 strains respectively (Fig 4).

## Impact of limit of quantification

For both strains, several measurements between the limit of detection and limit of quantification were recorded at the intracellular and extracellular level, mainly at the onset of infection

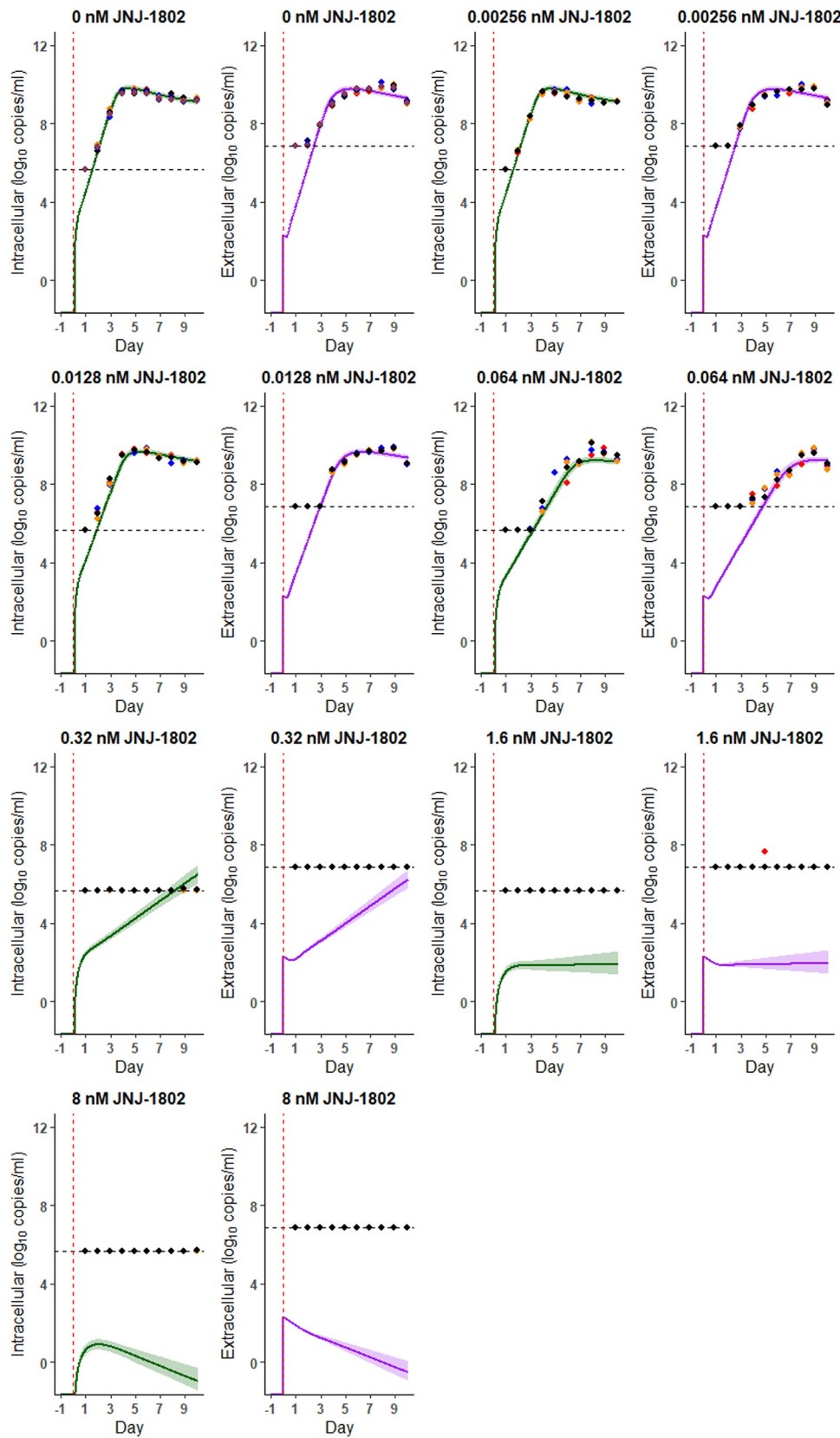

**Fig 3. Model Fits (DENV-2/RL strain).** Model fits for the measurements observed using the DENV-2/RL strain with no medium refresh and antiviral concentrations of 0 nM, $2.56 \times 10^{-03}$ nM, $1.28 \times 10^{-02}$ nM, $6.40 \times 10^{-02}$ nM, 0.32 nM, 1.6 nM, and 8 nM JNJ-1802. Coloured points represent the data from each well (6 wells for 0 nM, 4 wells for concentrations >0 nM), the vertical red line indicates the time the viral inoculum was added to each well and the horizontal dashed black line indicates the lower limit of detection. The modelled dynamics of the intracellular RNA virus are in green and those of the extracellular RNA virus are in purple; solid lines represent the median and the shading represents the 95% CrI. Here, the measurements below the limit of quantification (crosses in Fig 1) were included during model fitting. Measurements below the LOD were left-censored at the LOD during model fitting and we plot these measurements at the LOD for visual display. We assumed that the antiviral directly inhibits the transition process of infected cells to infectious (virion producing) cells, i.e., acts on τ.

and at days 8–10 p.i., most notably for concentrations of 1.6 nM or greater (crosses in Fig 1). Given that the quantification of measurements below the limit of quantification is considered less reliable compared to those at or above this limit, we tested the sensitivity of the results to left-censoring at the limit of quantification instead of left-censoring at the limit of detection during model fitting, using a probabilistic approach (Eq (11) below). This sensitivity analysis provided insight into the impact of these measurements on the parameter estimates and the concentration-effect profile of the antiviral. For both strains, the 95% CrI of the posterior estimates in the two scenarios largely overlapped (Table B in S1 Text). The most notable impact of left-censoring at the limit of quantification was on the concentration-effect profile of the antiviral as, for both strains, we obtained higher estimates of the $IC_{50}$ and Hill coefficient and estimated that maximum inhibition could be reached at a lower concentration when we left-censored at the limit of quantification (Table B and Figs H-J in S1 Text), corresponding to viral suppression for concentrations of 0.32 nM or greater. This is because for concentrations of 0.32 nM and above, all intracellular and extracellular DENV-2 RNA measurements were below the limit of quantification. Similarly, the effective reproduction number was estimated to reach 1 at a JNJ-1802 concentration of approximately 0.92 nM (95% CrI: 0.63 nM, 1.22 nM) and 0.66 nM (95% CrI: 0.55 nM, 0.75 nM) for the DENV-2/RL and DENV-2/16681 strains

**Table 1. Posterior parameter estimates.** Median posterior estimate and 95% credible interval (CrI) in brackets. Here the measurements below the limit of quantification (crosses in Fig 1) were included during model fitting, the well medium was not refreshed, and we assumed that the antiviral directly inhibits transition of infected cells to infectious (virion producing) cells, i.e., acts on τ.

| Parameter | Description | DENV-2/RL | DENV-2/16681 |
|---|---|---|---|
| β | Infection rate of target cells per virion (day$^{-1}$) | $3.18 \times 10^{-08}$ ($2.29 \times 10^{-08}$, $4.40 \times 10^{-08}$) | $1.32 \times 10^{-08}$ ($9.05 \times 10^{-09}$, $1.88 \times 10^{-08}$) |
| ω | Intracellular virus production rate per infectious cell (day$^{-1}$) | $3.17 \times 10^{6}$ ($2.32 \times 10^{6}$, $4.44 \times 10^{6}$) | $1.36 \times 10^{7}$ ($9.43 \times 10^{6}$, $1.89 \times 10^{7}$) |
| p | Proportion of intracellular RNA becoming extracellular virus | $5.90 \times 10^{-01}$ ($3.99 \times 10^{-01}$, $8.83 \times 10^{-01}$) | $2.28 \times 10^{-01}$ ($1.55 \times 10^{-01}$, $3.55 \times 10^{-01}$) |
| $IC_{50}$ | Concentration at which 50% of maximum effect is achieved (nM) | $1.23 \times 10^{-02}$ ($8.33 \times 10^{-03}$, $1.66 \times 10^{-02}$) | $1.28 \times 10^{-02}$ ($9.00 \times 10^{-03}$, $1.79 \times 10^{-02}$) |
| h | Hill coefficient | 1.28 (1.14, 1.42) | 1.17 (1.07, 1.31) |
| $R_0$ | Basic reproduction number | 277.01 (231.78, 332.66) | 191.10 (161.84, 232.76) |
| $\sigma_v$ | Residual error standard deviation (extracellular measurements) | 0.83 (0.78, 0.88) | |
| $\sigma_X$ | Residual error standard deviation (intracellular measurements) | 0.91 (0.86, 0.97) | |
| L | Log-likelihood | -1619.87 (-1627.66, -1614.25) | |
| DIC | Deviance Information Criterion | 3,254 | |

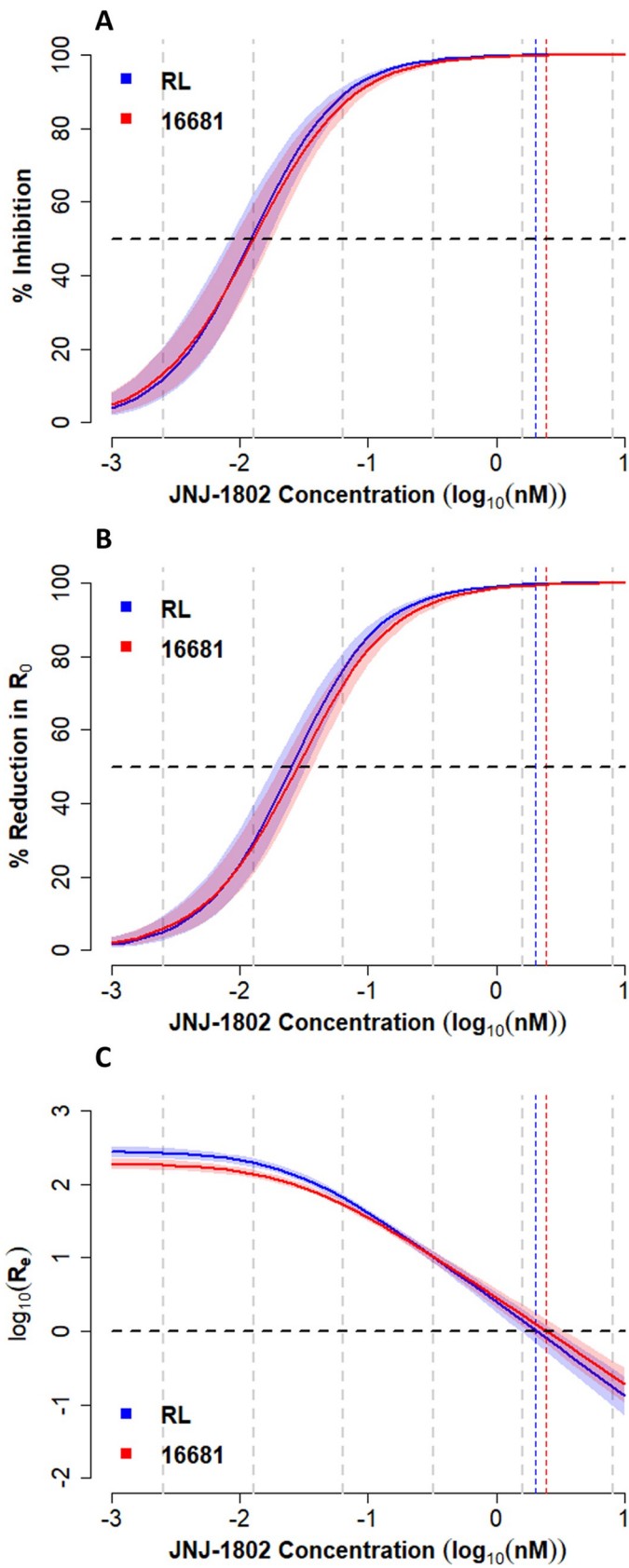

**Fig 4. Effect of antiviral.** Estimated inhibition percentage (A), percentage reduction in the basic reproduction number $R_0$ (B) and effective reproduction number $R_e$ (C) as a function of concentration of JNJ-1802, for the experiments without well medium refresh on day 4 for the RL and 16681 DENV-2 strains. Estimates were calculated by substituting 1,000 parameter values sampled from the posterior distribution into *Eq*s (1) and (6). Solid lines represent the median, the shading represents the 95% CrI, dotted grey vertical lines indicate the concentrations tested in the in-vitro experiments, and the dotted blue and red vertical lines indicate the median concentration such the $R_e$ = 1. The dotted black horizontal lines indicate the threshold for a 50% reduction (A,B), and when $R_e$ = 1 (C). Here, measurements below the limit of quantification (crosses in Fig 1) were included during model fitting and we assumed that the antiviral directly inhibits the transition process of infected cells to infectious (virion producing) cells, i.e., acts on τ.

respectively (Fig J in S1 Text), which is lower compared to when left-censoring was assumed at the limit of detection.

## Impact of latent period and JNJ-1802 effect

Viral kinetic data obtained in a clinical setting often do not have enough granularity to allow quantification of the transition from infected to infectious cells. Therefore, a simplification of the viral kinetic model is often required by removing the latent period allowing for infected cells to become infectious, and with the drug directly inhibiting production of virions [51–55].

In a first sensitivity analysis, the drug effect was assumed on intracellular RNA production, instead of the transition from infected to infectious cells, while retaining the latent period (Section 3 in S1 Text). Compared to the original model, for both strains, we obtained comparable model fits and parameter estimates (Figs N and O and Table C in S1 Text), and found similar between-strain differences with respect to concentration-effect curves (Figs 4 and P in S1 Text). The main differences observed were in the underlying cell dynamics at intermediate concentrations of JNJ-1802 (e.g., 0.064 nM) where we estimated a higher number of infectious cells compared with the original model owing to the different assumed mechanism of drug action (Figs E,F,L, and M in S1 Text). For all experiments, we estimated a marginally higher median $IC_{50}$ when the antiviral was assumed to inhibit intracellular RNA production rather than inhibiting the transition of infected cells to infectious cells (Table C in S1 Text). In addition, a marginally lower median concentration (1.65 nM and 1.99nM for RL and 16681 strain, respectively) was needed to bring the effective reproduction number below 1 when the antiviral was assumed to act on the intracellular RNA production (Fig P in S1 Text).

In a subsequent sensitivity analysis, the latent period was removed by excluding the compartment of infected cells, allowing target cells to become infectious immediately, as implemented in various viral kinetics models to describe *in-vivo* data [46,51–53] with a direct drug effect on intracellular RNA production. Compared to the original model, for both strains, we estimated a marginally higher median $IC_{50}$ and observed an approximately 70% reduction in the estimated basic reproduction number (Table C in S1 Text). However, for both strains, broadly similar concentration-effect profiles were obtained, with a slighter lower median concentration (1.41 nM and 1.62 nM for RL and 16681 strain, respectively) required to bring the effective reproduction number below 1 when the latent period was removed (Fig P in S1 Text).

Intracellular data collected on day 4 for experiments using the DENV-2 16681 strain and JNJ-1802 concentrations of 0 nM, $2.56 \times 10^{-03}$ nM, $1.28 \times 10^{-02}$ nM, and $6.40 \times 10^{-02}$ nM was excluded from the final model fit because this data was inconsistent with the corresponding extracellular data (Fig 1). In addition, measurements with $> 1$ $\log_{10}$ difference from the corresponding median during model fitting were also excluded because they were considered outlying (Fig 1). We tested the sensitivity of our results to the exclusion of this data and found that, across all scenarios, the 95% credible intervals for our posterior estimates of the $IC_{50}$, Hill coefficient and $R_0$ were largely overlapping (Tables F and G in S1 Text) and thus we obtain consistent estimates of the *in-vitro* prophylactic inhibitory effect profile of JNJ-1802.

The model selected based on the Deviance Information Criterion (DIC)[57] was the one where we assumed a latent period with the antiviral to act on the transition from infected to infectious cells, in line with the mechanism of action of JNJ-1802 in preventing the NS3-NS4B interaction required for viral replication (Table C in S1 Text).

## Discussion

JNJ-1802 is one of the most advanced candidate dengue antivirals under development. While the NS3-NS4B interaction blocking mechanism of action of this novel antiviral was previously established [43], robustly quantifying the effect of the antiviral on viral dynamics both *in-vivo* and across different host systems is critical to developing an understanding of the potential impact of antiviral treatment for infection and disease control. The first step in this process is examining the antiviral effect *in-vitro*, where the host immune system does not play a role in viral elimination, and this study provides the first model-based estimates of the *in-vitro* prophylactic inhibitory effect of JNJ-1802 based on both intra- and extracellular viral RNA profiles. Unlike regression-based statistical approaches to compare viral RNA measurements collected at a single timepoint [38,41], this approach allows us to consider the full temporal profile of viral RNA thereby enabling us to obtain an estimate of the basic reproduction number ($\mathcal{R}_0$), to quantify the impact of JNJ-1802 on this key characteristic, and quantify the *in-vitro* prophylactic inhibitory effect profile.

In this context, this study enhances our understanding of the viral kinetics of dengue infection *in-vitro* (in the absence of an antiviral molecule) and sheds some light on the within-serotype variation among DENV-2 strains. For the DENV-2/RL and DENV-2/16681 strains, we estimated a median basic reproduction number of 277 and 191, respectively. Hence, the model suggests that the DENV-2/RL strain had a fitness advantage in Vero cells compared with the DENV-2/16681 strain. To the best of our knowledge, these are the first estimates of DENV viral fitness, as measured by the *in-vitro* $R_0$, which not only provide important insight into the transmissibility of these strains, but also allow to quantify the reduction in $R_0$ required for viral suppression ($R_e < 1$). Here we estimate an average reduction in $R_0$ of 99.6% and 99.4% is required for DENV-2/RL and DENV-2/16681 strains, respectively.

We found that the effect of JNJ-1802 on viral replication was strongly concentration-dependent, with near-complete inhibition (>99%) achieved at concentrations of $\geq 1.6$ nM, and similar $IC_{50}$ estimates for the DENV-2/RL strain ($IC_{50}$: $1.23 \times 10^{-02}$ nM (95% CrI: $8.33 \times 10^{-03}$ nM, $1.66 \times 10^{-02}$ nM), Hill coefficient (h): 1.28 (95% CrI: 1.14,1.42)) and the DENV-2/16681 strain ($IC_{50}$: $1.28 \times 10^{-02}$ nM (95% CrI: $9.00 \times 10^{-03}$ nM, $1.79 \times 10^{-02}$ nM, h: 1.17 (95% CrI: 1.07,1.31)). In comparison, as presented by Goethals et al.[43], the estimates obtained from single timepoint experiments in Vero cells using regression-based techniques gave an average $EC_{50}$ of $5.9 \times 10^{-02}$ nM and $6.2 \times 10^{-02}$ for the DENV-2/16681 and DENV-2/RL strains, respectively. We estimated a JNJ-1802 concentration of greater than approximately 2.02 nM and 2.43 nM for the DENV-2/RL and DENV-2/16681 strains respectively would be required to bring the *in-vitro* effective reproduction number below 1. These concentrations are considerably below the estimated cellular toxicity ($CC_{50}$) levels for JNJ-1802 in Vero cells (RL: >2260 ± 350 nM, 16681: 2610 ± 470 nM)[43].

The more conservative approach of assuming RNA measurements were left-censored at the limit of quantification, rather than the limit of detection, provided similar viral kinetic parameter estimates, but altered the estimated concentration-effect profile of JNJ-1802, with viral suppression and maximum inhibition achieved at lower concentrations. However, the extent to which measurements below the limit of quantification, but above the limit of detection, were representative of the true dynamics or were affected by limitations of RNA quantification and

increased measurement error, cannot be determined. Our results highlight the importance of assessing the robustness of drug effect estimates regarding assumptions on the reliability of viral RNA measurements below the limit of quantification.

An additional limitation of our study is that several model parameters were fixed, including those governing the underlying cell dynamics. We also did not have an exact measurement of infectious virus at the start of infection (which is below the limit of detection), and thus we used the MOI to set the initial value of extracellular RNA concentration in our model ($V_0$) and assumed this value was the same for both strains. Our estimates of $R_0$ are sensitive to the value of $V_0$ chosen, with higher values of $V_0$ resulting in lower estimates of $R_0$ (Tables D and E in S1 Text). Nonetheless, our estimates of the $IC_{50}$ and Hill co-efficient were not sensitive to the value of $V_0$ chosen (Tables D and E in S1 Text) and hence our characterisation of the *in-vitro* prophylactic inhibitory effect profile of JNJ-1802 is robust to these assumptions.

One of the key advantages provided by these *in-vitro* experiments, in conjunction with the rich longitudinal data generated, is the detailed characterization of the viral kinetics in cells prior and during an infection. To this end, it was possible to identify a transition process of infected cells to infectious cells (latent period) with a direct effect of JNJ-1802 on this transition. This is in line with the mechanism of action of JNJ-1802, that blocks the NS3-NS4B interaction and hence formation of the replication complex, therefore blocking viral RNA production and virus replication [43].

Viral kinetic models calibrated against *in vivo* challenge data (e.g., in mice, non-human primates, humans) often do not contain enough information to identify the duration of this latent period [51–53,58]. In a sensitivity analysis, it was shown that comparable parameter estimates and concentration-effect profiles could be obtained in absence of this latent period, although this model was less preferred based on DIC compared to the model including a latent period (Table C in S1 Text).

In addition, our results also indicate that refreshing the well medium on day 4 did not affect the viability of the Vero cells and did not substantially alter the viral kinetics nor the concentration-effect profiles. Nonetheless, consistency in the experimental design is beneficial for comparison purposes and to robustly explore differences between DENV serotypes and strains.

Looking ahead, the results of experimental challenge studies recently conducted in mice and non-human primates will provide both further insight into the *in-vivo* inhibitory effect of JNJ-1802 across host species [43], the interplay between the virus, the host, JNJ-1802 and its dosing scheme and the generalisability of the models developed in this study. We anticipate that the model of *in-vitro* viral kinetics developed here will form the basis for developing *in-vivo* viral kinetics models, which may include more complex interactions such as the host immune response to infection, and differences in dengue pathogenesis across host species. In addition, the modelling framework developed in this study offers the potential to investigate further how viral kinetics, JNJ-1802 effect, and between-serotype differences can be translated across different host systems and to inform selection of dosing regimens. Whilst caution is required in the extrapolation of results from *in-vitro* studies to different hosts, as the translation needs to be explored further with actual data from in-vitro studies and human clinical studies that are currently ongoing, this work demonstrates that longitudinal viral kinetics observations can be reconstructed using mathematical models. Hence, if this translation could be established, it is our hypothesis that it may be possible to use models fitted to early *in-vitro* data as a tool to explore the potential impact of a drug candidate across species, thus potentially informing the design of *in-vivo* and human studies.

To date, there remains an urgent need for the development of effective, specific antiviral treatment for dengue and this study shows that, for two different strains of the DENV-2

serotype, JNJ-1802 is both fast-acting and successful at inhibiting dengue virus replication *in-vitro*. The work presented here is a first step in the pathway of quantifying the potential of JNJ-1802 as a promising and effective antiviral candidate against dengue infection and disease.

## Methods

### Experimental design

**Cells.** Vero cells (African green monkey kidney cells, European Collection of Cell Cultures [ECACC]) were cultured in Eagle's minimum essential medium (MEM), with phenol red (Gibco, Thermo Fisher Scientific), supplemented with 10% fetal bovine serum (FBS; Bio West), 0.04% gentamycin (Gibco; concentration stock solution: 50 mg/mL), and 2 mM Ala-glutamine (Sigma). In the kinetics experiments, Vero cells were cultured in MEM medium without phenol red (Gibco, Thermo Fisher Scientific) containing 2% FBS.

**Virus.** Lab-adapted strain DENV-2/16681 was produced by transfection of the *in-vitro* transcribed RNA of plasmid pFK-DVs into Huh7 cells. This plasmid encodes for the full-length DENV-2/16681. Plasmid pFK-DVs was obtained by insertion of a synthetic copy of the full-length genomic sequence of DENV-2 strain 16681 (GenBank Accession NC_001474) into the low-copy plasmid vector pFK. Moreover, the parental vector pFK was modified by insertion of the SP6 promoter upstream of the DENV 5′-NTR to allow synthesis of authentic viral RNA by *in-vitro* transcription. This plasmid was licensed from Prof. R. Bartenschlager [59]. Briefly, the plasmid pFK-DVs was linearized by *Xba*I (Thermo Fisher Scientific) and the purified DNA was *in-vitro* transcribed using the mMESSAGE mMACHINE SP6 kit (Thermo Fisher Scientific). Virus stocks were made by transfection of the purified *in-vitro*-transcribed RNA into Huh7 cells by electroporation, using a Gene Pulser II Electroporation System (Bio-Rad Laboratories). The cells were incubated for 24 hours at 37˚C and 5% $CO_2$, followed by incubation for 3 days at 30˚C and 5% $CO_2$. Finally, cell culture supernatant was harvested, and the resulting virus stock was stored at -80˚C until further use. New batches of DENV-2 were obtained by infecting C6/36 cells with the respective virus. The DENV-2/RL strain (GenBank accession MW741553) was recultured by infecting C6/36 cells with the respective virus [42]. For both strains, virus stocks were titrated and the 50% endpoint titer, expressed in $CCID_{50}$ (50% cell culture infectious dose) per mL, was determined according to the Reed Muench formula [60].

**Measurement of the viral kinetics of DENV-2/RL and DENV-2/16681 in Vero cells.** A two-step duplex and simplex RT-qPCR assay was performed to determine the amount of respectively intracellular and extracellular viral RNA of the DENV-2/RL or DENV-2/16681 in presence and absence of JNJ-1802.

An overview of the timeline for each experiment is provided in Table 2 below. Briefly, for each strain and each concentration of JNJ-1802 tested, 16 plates were seeded with Vero cells in MEM medium (2% FBS) at a density of 5,000 cells/well in 96-well plates. Six wells were used to examine viral kinetics in the absence of JNJ-1802, and four wells were used to examine the viral kinetics in the presence of JNJ-1802 (concentration range: 0.00256 nM—8 nM with four wells per concentration). For treatment wells, JNJ-1802 was added to wells before cells were seeded. 24 Hours after seeding, the cells were infected with DENV (MOI 0.01) and incubated at 37˚C. For 10 of these 16 plates, the medium was not refreshed during the experiment and each day for 10 days, one plate of these 10 plates was further processed. For the other 6 plates, the medium was refreshed at day 4 post infection, and from day 5 onwards, one plate was harvested each day until day 10. Thus, for each strain, viral RNA measurements were obtained from a total of 480 wells.

**Table 2. Experiment Design.** Overview of the experiment for each DENV-2 strain. For each strain and concentration of JNJ-1802 tested, 16 plates were seeded with Vero cells at a density of 5,000 cells/well. Six wells were used to examine viral kinetics in the absence of JNJ-1802, and four wells were used to examine the viral kinetics in the presence of JNJ-1802 (concentration range: 0.00256 nM-8 nM .24 Hours after seeding, the cells in each well were infected with DENV-2). For 10 of the 16 plates, the medium was not refreshed during the experiment and each day for 10 days, one of these plates was further processed. For the other 6 plates, the medium was refreshed at day 4 post infection and from day 5 onwards, one plate was harvested each day until day 10.

| | | *Seed Cells* | | | | | | | | | | | |
|---|---|---|---|---|---|---|---|---|---|---|---|---|---|
| **Day p.i.** | | -1 | 0 | 1 | 2 | 3 | 4 | 5 | 6 | 7 | 8 | 9 | 10 |
| **Plate Harvested** | No Refresh | | | Plate 1 | Plate 2 | Plate 3 | 4 | Plate 5 | Plate 6 | Plate 7 | Plate 8 | Plate 9 | Plate 10 |
| | With Refresh | | | | | | *Refresh* | Plate 11 | Plate 12 | Plate 13 | Plate 14 | Plate 15 | Plate 16 |
| **No. of wells per plate** | 0 nM | | | 6 | 6 | 6 | 6 | 6 | 6 | 6 | 6 | 6 | 6 |
| | 0.00256 nM | | | 4 | 4 | 4 | 4 | 4 | 4 | 4 | 4 | 4 | 4 |
| | 0.0128 nM | | | 4 | 4 | 4 | 4 | 4 | 4 | 4 | 4 | 4 | 4 |
| | 0.064 nM | | | 4 | 4 | 4 | 4 | 4 | 4 | 4 | 4 | 4 | 4 |
| | 0.32 nM | | | 4 | 4 | 4 | 4 | 4 | 4 | 4 | 4 | 4 | 4 |
| | 1.6 nM | | | 4 | 4 | 4 | 4 | 4 | 4 | 4 | 4 | 4 | 4 |
| | 8 nM | | | 4 | 4 | 4 | 4 | 4 | 4 | 4 | 4 | 4 | 4 |

As described above, the supernatant was collected from each plate and stored in 96-Well V-bottom plates at -80˚C. Intracellular RNA was measured by washing the adherent cells of the plates without supernatant with cold PBS and plates were incubated at -80˚C for 24 hours. After 24 hours the cells were lysed with Cells-to-$C_T$ Bulk Lysis Reagents kit (Thermo Fisher Scientific) and the cell lysates were used to prepare cDNA (using Expand Reverse Transcriptase) of the target sequences, the 3'-untranslated region (UTR) of DENV, and the cellular housekeeping reference gene β-actin. Furthermore, extracellular RNA was measured by extracting the viral RNA from the collected supernatant and subsequent cDNA preparation as described above. Subsequently, a duplex (intracellular RNA) or simplex (extracellular RNA) real-time PCR was performed on a Lightcycler480 instrument (Roche) at the following conditions: 10 minutes at 95˚C, followed by 40 cycles of 10 seconds at 95˚C, 1 minute at 60˚C.

For the intracellular RNA quantification, the limit of detection of the RT-qPCR assay was 5.6 $\log_{10}$ copies/ml and 5.0 $\log_{10}$ copies/ml for the DENV-2/RL and DENV-2/16681 strains respectively, and the limit of quantification was 6.6 $\log_{10}$ copies/ml and 5.6 $\log_{10}$ copies/ml for the DENV-2/RL and DENV-2/16681 strains respectively, calculated using an RNA standard curve within the same experiment. For the extracellular RNA quantification, the limit of detection and the limit of quantification of the RT-qPCR assay was 6.9 $\log_{10}$ copies/ml for both the DENV-2/RL and DENV-2/16681 strain, calculated using an RNA standard curve from a different experiment.

## Mathematical model

To estimate the JNJ-1802 antiviral effect on the dynamics of a dengue infection in Vero cells, we developed a mechanistic model of the within-host dynamics of dengue *in-vitro*, by modifying a model developed by Clapham *et al.*[53] to explore the within-host dynamics of dengue infection *in-vivo*. In addition to not including a host immune response to infection, our model differs to that developed by Clapham *et al.*[53] by describing a more detailed representation of the pathway from target cell infection to extracellular viral RNA production.

We use a deterministic compartmental model to describe the dynamics in each well as we aim to describe average behaviour at the population (rather than individual cell) level and to estimate the average rates at which events (e.g., infection of target cells) typically occur in the population. The model schematic is provided in Fig 5 and describes the *in-vitro* transmission

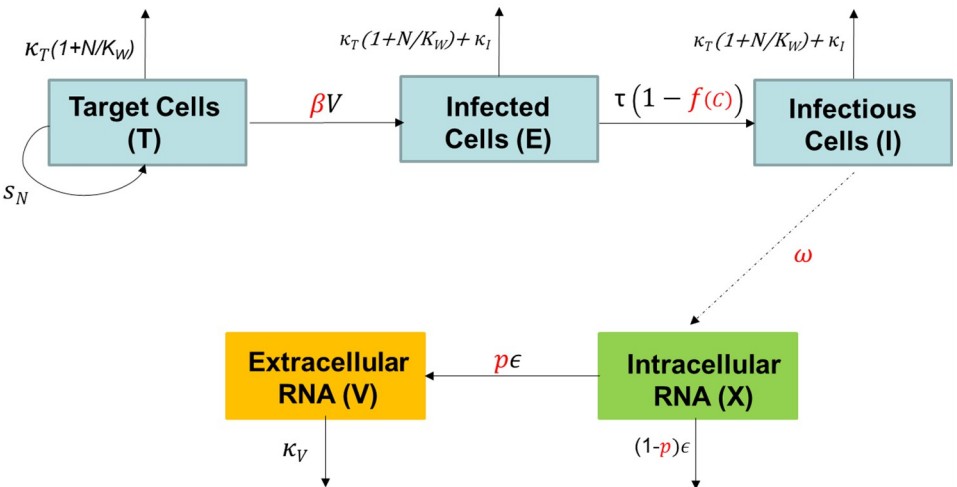

**Fig 5. Model Schematic.** Target cells are infected at a rate β per virion. Following a latent period of $1/\tau$ days on average (in absence of antiviral drug), infectious cells $I$ synthesize intracellular viral RNA ($X$) at a net production rate $\omega$, a proportion p of which then gets secreted as extracellular viral RNA at rate $\varepsilon$. Extracellular viral RNA decays at rate $\kappa_V$. Target cells replicate at rate $s_N$; target cells and infected cells have a mean lifespan of $1/\kappa_T$ and $1/(\kappa_T + \kappa_I)$ days, respectively. Individual wells have a carrying capacity $K_W$. The antiviral molecule acts on the transition process of infected to infectious (virion producing) cells ($\tau$), following a concentration-dependent function f(C). The parameters in red are estimated, those in black are fixed to the values reported in Table *2*.

dynamics of dengue virus in each well following viral inoculation. Table 2 provides a summary of the parameters used in the model.

We assume that target cells $T$ replicate at a rate $s_N$ and have a mean lifespan of $1/k_T$ days. However, as resources in individual wells were limited, we assume there is a limit to the size of the cell population each well can accommodate, and that regulation of the cell population is achieved via density-dependent mortality whereby mortality increases as the total number of cells reaches the carrying capacity ($K_w$) of the well.

Target cells become infected at a second-order rate constant, $\beta V$. Therefore, the infection rate is proportional to the abundance of both $T$ and $V$, the extracellular viral RNA concentration in the well (copies/mL). Following a latent period with a mean duration of $1/\tau$ days (in absence of antiviral drug), infectious cells $I$ synthesize intracellular viral RNA ($X$) at a net production rate $\omega$, a proportion $p$ of which then gets secreted as extracellular viral RNA at a rate $\varepsilon$. Finally, extracellular RNA is assumed to decay at a rate $k_V$. We assume that infection increases the decay rate of cells by a fixed value, $k_I$.

We assume that JNJ-1802 directly inhibits the transition process of infected cells to infectious (i.e., virion producing) cells ($\tau$). As the drug concentration remained constant throughout the course of each experiment, we assume that the magnitude of the effect $f(C)$ also remained constant throughout the course of each experiment. We capture the relationship between the antiviral drug concentration $C$ (nM) and the magnitude of the inhibitory effect $f(C)$ using a Hill function [61,62] as follows:

$$f(C) = \frac{I_{max}C^h}{(IC_{50})^h + C^h} \in [0, 1] \tag{1}$$

where $I_{max} \in [0,1]$ denotes the maximum inhibitory effect of the antiviral, $IC_{50}$ denotes the concentration which achieves 50% of the maximum inhibitory effect and $h$ denotes the Hill coefficient.

The dynamics of the cell population in each well are described by the following set of deterministic equations:

$$\frac{dT}{dt} = s_N T - k_T\left(1 + \frac{N}{K_w}\right)T - \beta V T$$

$$\frac{dE}{dt} = \beta V T - k_T\left(1 + \frac{N}{K_w}\right)E - k_I E - \tau(1 - f(C))E \tag{2}$$

$$\frac{dI}{dt} = \tau(1 - f(C))E - k_T\left(1 + \frac{N}{K_w}\right)I - k_I I$$

$$\frac{dX}{dt} = \omega I - \epsilon X$$

$$\frac{dV}{dt} = p\epsilon X - k_V V$$

where $N = T+E+I$. The carrying capacity of the well $K_w$ is obtained by considering the equilibrium cell population $N^*$ in each well under disease-free conditions ($E = I = V = 0$). For an equilibrium population $N^*$ we have that $s_N - k_T(1 + N^*/K_w) = 0$, and hence

$$K_w = \frac{k_T N^*}{s_N - k_T} \tag{3}$$

The basic reproduction number ($\mathcal{R}_0$) for this model is defined as the mean number of infected cells produced by each infected cell over its lifespan under disease-free conditions (at the start of infection, time t*) and without therapeutic intervention and is given by:

$$\mathcal{R}_0 = \left(\frac{\omega p \,\beta T^*}{k_I^* k_V}\right)\left(\frac{\tau}{k_I^* + \tau}\right) \tag{4}$$

where $k_I^* = k_I + k_T\left(1 + \frac{T^*}{K_w}\right)$ and

$$T^* = T(t^*) = \frac{K_w(s_N - k_T)T_0}{k_T T_0 + [K_w(s_N - k_T) - k_T T_0]e^{(-(s_N - k_T)t^*)}} \tag{5}$$

denotes the target cell population at the start of infection (time t*). A full derivation of $\mathcal{R}_0$ is provided in Section 6 in S1 Text.

We define the effective reproduction number ($\mathcal{R}_e$) as follows:

$$\mathcal{R}_e = \left(\frac{\omega p \,\beta T^*}{k_I^* k_V}\right)\left(\frac{\tau(1 - f(C))}{k_I^* + \tau(1 - f(C))}\right) \tag{6}$$

where $f(C)$ is defined as in Eq (1) above.

For a given concentration $C$, we estimate the percentage reduction in the basic reproduction number $e(C)$ as follows:

$$e(C) = 1 - \frac{\mathcal{R}_e}{\mathcal{R}_0} = 1 - \left( \frac{\tau(1 - f(C))}{k_I^* + \tau(1 - f(C))} \right) \left( \frac{k_I^* + \tau}{\tau} \right)$$

$$= \frac{(k_I^* + \tau(1 - f(C))) - (k_I^* + \tau)(1 - f(C))}{k_I^* + \tau(1 - f(C))} \tag{7}$$

$$= \frac{(k_I^* + \tau) - \tau f(C) - (k_I^* + \tau) + (k_I^* + \tau)f(C)}{k_I^* + \tau(1 - f(C))}$$

$$= \frac{k_I^* f(C)}{(k_I^* + \tau) - \tau f(C)}$$

Substituting Eq (1) into Eq (7) above, we have:

$$e(C) = \frac{k_I^* I_{max} C^h}{(k_I^* + \tau)((IC_{50})^h + C^h) - \tau I_{max} C^h}$$

$$= \frac{k_I^* I_{max} C^h}{(k_I^* + \tau)(IC_{50})^h + (k_I^* + \tau - \tau I_{max})C^h} \tag{8}$$

$$= \left( \frac{k_I^* I_{max}}{k_I^* + \tau(1 - I_{max})} \right) \frac{C^h}{\frac{(k_I^* + \tau)(IC_{50})^h}{k_I^* + \tau(1 - I_{max})} + C^h}$$

Thus we have:

$$e(C) = \frac{I_{max}^* C^h}{(IC_{50}^*)^h + C^h} \tag{9}$$

where

$$I_{max}^* = \frac{k_I^* I_{max}}{k_I^* + \tau(1 - I_{max})} \ and \ IC_{50}^* = \left( \frac{k_I^* + \tau}{k_I^* + \tau(1 - I_{max})} \right)^{\frac{1}{h}} IC_{50}$$

When $I_{max} = 1$, we have:

$$I_{max}^* = 1 \ and \ IC_{50}^* = \left( 1 + \frac{\tau}{k_I^*} \right)^{1/h} IC_{50} \tag{10}$$

Using Eqs (1) and (6) above, we have that a concentration $C_{crit}$ with

$$C_{crit} \geq \left( \frac{z_1}{I_{max} z_2 - z_1} \right)^{1/h} IC_{50} \tag{11}$$

where

$$z_1 = \omega p \ \beta T^* \tau - k_I^* k_V (k_I^* + \tau), z_2 = \omega p \ \beta T^* \tau - k_I^* k_V \tau$$

is required to bring the effective reproduction number ($\mathcal{R}_e$) below 1.

## Parameter estimation

For each strain and experiment type (with/without medium refresh), we fitted the model simultaneously to the observed intracellular and extracellular viral RNA measurements collected in the experiments using the Metropolis-Hastings algorithm, a Markov chain Monte Carlo (MCMC) method[63].

We excluded measurements with $> 1 \log_{10}$ difference from the corresponding median value (26 measurements in total, red points in Fig 1) and the intracellular data collected on day 4 for experiments using the DENV-2/16681 dengue strain and antiviral concentrations of 0 nM, $2.56 \times 10^{-03}$ nM, $1.28 \times 10^{-02}$ nM, and $6.40 \times 10^{-02}$ nM, as these were inconsistent with the corresponding extracellular data (18 measurements in total, red squares in Fig 1).

For an individual experiment, following Clapham *et al.*[53], let $X_{ijk}$ and $V_{ijk}$ denote the observed $\log_{10}$ intra- and extracellular viral RNA measurements in the $j^{\text{th}}$ well of dosing arm $i$ on day $k$, with the corresponding model predicted values represented by $\hat{X}_{ijk}$ and $\hat{V}_{ijk}$. The residual error between the observed and predicted values was assumed to be normally distributed with variance $\sigma^2$. The intra- and extracellular viral RNA measurements below the limit of detection (LOD) $LOD_X = \log_{10}(LOD_X)$ and $LOD_V = \log_{10}(LOD_V)$ were accounted for in the likelihood using the cumulative distribution function, with the likelihood $\mathbb{L}$ for an individual experiment given by:

$$\mathrm{L} = \prod_{Y=\{X,V\}} \prod_{i=1}^{I} \prod_{j=1}^{J} \prod_{k=1}^{K} \phi\left((Y_{ijk}|\hat{Y}_{ijk}, \sigma_Y^2)\right)^{c_{Yijk}} \Phi\left((LOD_l|\hat{Y}_{ijk}, \sigma_Y^2)\right)^{1-c_{Yijk}} \tag{12}$$

where $\phi$ and $\Phi$ denote the probability density function (pdf) and cumulative distribution function (cdf) of the Normal distribution respectively, the indicator function $c_{Yijk}$ is 1 if the observed value was above the limit of detection ($Y_{ijk} > LOD_Y$) and 0 otherwise, $Y$ denotes the variable of interest, $I$ denotes the total number of dosing arms, $J$ denotes the total number of wells in each dosing arm, and $K$ denotes the total number of days. Thus, left-censored data were accounted for using a probabilistic approach rather than using imputation methods.

We estimated 7 parameters in total ($\beta$, $\omega$, $p$, $IC_{50}$, $h$, $\sigma_X$, $\sigma_V$). $\beta$, $\omega$, $p$, $IC_{50}$, and $h$ were estimated at the individual experiment level and $\sigma_X$, and $\sigma_V$ were estimated globally. MCMC was run for 500,000 iterations, parameters were estimated on a $\log_{10}$ scale as a single block using uninformative uniform prior distributions and a multivariate Normal proposal distribution. Details on the range used for the prior distribution assumed for the estimated model parameters are provided in Table 3.

The carrying capacity of individual wells was set based on estimates of the doubling time of Vero cell cultures and the level of confluency observed after seeding. At the time of infection (24 hours post-seeding), cells had reached approximately 70% confluency. It has been observed elsewhere that Vero cell cultures double approximately every 24 hours [64], and thus, given that each well was seeded with 5,000 cells, we estimate that 70% confluency corresponds to approximately 10,000 cells. Assuming 100% confluency corresponds to the carrying capacity of an individual well, this gives us an approximate equilibrium population size of 14,000 cells.

We assumed an average time between a cell becoming infected and infectious (virion producing) of 12 hours based on work by Matsumura *et al.* examining the development of DENV in Vero cells[65]. Results from Matsumura *et al.*[65] also show that mature virions are present 24–36 hours p.i., and thus we set the average intracellular RNA secretion rate to 2 day$^{-1}$ to give an approximate duration of 24 hours between a cell becoming infected and extracellular RNA being secreted.

**Table 3. Model Parameters.** Parameters used in the mathematical model, including notation, description, values, reference publications and prior distributions used for estimated parameters (U = uniform, lower and upper bounds are provided within brackets).

| Symbol | Parameter Description | Parameter Value | Reference | Prior |
|---|---|---|---|---|
| $s_N$ | Cell growth rate (day$^{-1}$) | 1.7 (set to give doubling time ~1 day) | [64] | n/a |
| $k_T$ | Target cell mortality rate (day$^{-1}$) | 0.1 (set to give doubling time ~1 day) | [64] | n/a |
| $k_I$ | Increase in cell mortality rate due to infection (day$^{-1}$) | 0.04 (set to give infected cell mortality ~ 7 days) | [52] | n/a |
| $1/\tau$ | Latent period (day$^{-1}$) | 0.5 | [65] | n/a |
| $\beta$ | Target cell infection rate (per virion day$^{-1}$) | Estimated | n/a | U $[10^{-12}, 10^{-2}]$ |
| $\omega$ | Intracellular RNA synthesis rate (per infectious cell, day$^{-1}$) | Estimated | n/a | U$[0, 10^{10}]$ |
| $\varepsilon$ | Intracellular RNA secretion rate (day$^{-1}$) | 2 | [65] | n/a |
| $k_V$ | Extracellular RNA decay rate (day$^{-1}$) | 1 | - | n/a |
| p | Proportion of intracellular RNA becoming extracellular RNA | Estimated | n/a | U$[0,1]$ |
| $K_w$ | Carrying capacity of individual well | Assigned to give an equilibrium cell population of 14,000 cells | n/a | n/a |
| $E_{max}$ | Maximum effect of antiviral | 1 | n/a | n/a |
| $IC_{50}$ | Concentration which achieves 50% of maximum effect | Estimated | n/a | U$[0,1]$ |
| $h$ | Hill coefficient | Estimated | n/a | U$[0,10]$ |
| $V_0$ | Initial viral inoculum (copies per ml) | 200 | n/a | n/a |
| $\sigma_X$ | Residual error standard deviation (Intracellular measurements on log$_{10}$ scale) | Estimated | n/a | U$[0,10]$ |
| $\sigma_V$ | Residual error standard deviation (Extracellular measurements on log$_{10}$ scale) | Estimated | n/a | U$[0,10]$ |

The 95% Credible Intervals (CrI) around the central estimates were generated using 1,000 parameter sets sampled from the posterior distribution. The viral dynamics and the concentration-effect curves were obtained by substituting these parameter sets into the model equations and using Eq 4, respectively.

## Alternative mode of JNJ-1802 antiviral effect

We tested an alternative mode of drug effect and assumed that by targeting the NS3-NS4B interaction, the antiviral inhibits intracellular viral RNA production ($\omega$) rather than directly inhibiting the transition process of infected cells to infectious cells, as assumed in the main model. The flow diagram and the corresponding equations of this alternative model are provided in Section 3 in S1 Text. Model selection was based on the Deviance Information Criterion (DIC)[57].

## Supporting information

**S1 Data. Viral Kinetics Data.**
(XLSX)

**S1 Text. Fig A. Variation in viral RNA measurements.** Co-efficient of variation (CV) for the log$_{10}$ viral RNA measurement obtained from experimental infection studies in Vero cells, disaggregated by measurement type, JNJ-1802 concentration and DENV-2 strain. CV is the ratio of the standard deviation to the mean. **Fig B. Model Fits (DENV-2/16681 strain, with refresh).** Model fits for the measurements observed using the DENV-2/16681 strain with medium refresh on day 4 and antiviral concentrations of 0 nM, 2.56x10$^{-03}$ nM, 1.28x10$^{-02}$ nM, 6.40x10$^{-02}$ nM, 0.32 nM, 1.6 nM, and 8 nM. Coloured points represent the data from each well (6 wells for 0 nM, 4 wells for concentrations >0 nM), the vertical red line indicates the time

the viral inoculum was added to each well and the horizontal dashed black line indicates the limit of detection. The modelled dynamics of the intracellular RNA virus are in green and those of the extracellular RNA virus are in purple; solid lines represent the median and the shading represents the 95% CrI. Viral suppression is observed for concentrations 1.6 nM and 8 nM. Here, measurements below the limit of quantification (crosses in Fig 1) were included during model fitting. Measurements below the LOD were left censored at the LOD during model fitting and were plotted at the LOD for visual display. We assumed that the antiviral directly inhibits the transition process of infected cells to infectious (virion producing) cells, i.e., acts on $\tau$. **Fig C. Model Fits (DENV-2/RL strain, with refresh).** Model fits for the measurements observed using the DENV-2/RL strain with medium refresh on day 4 and antiviral concentrations of 0 nM, $2.56 \times 10^{-03}$ nM, $1.28 \times 10^{-02}$ nM, $6.40 \times 10^{-02}$ nM, 0.32 nM, 1.6 nM, and 8 nM. Coloured points represent the data from each well (6 wells for 0nM, 4 wells for concentrations >0nM), the vertical red line indicates the time the viral inoculum was added to each well and the horizontal dashed black line indicates the limit of detection. The modelled dynamics of the intracellular RNA virus are in green and those of the extracellular RNA virus are in purple; solid lines represent the median and the shading represents the 95% CrI. Viral suppression is observed for concentrations 1.6 nM and 8 nM. Here, measurements below the limit of quantification (crosses in Fig 1) were included during model fitting. Measurements below the LOD were left censored at the LOD during model fitting and were plotted at the LOD for visual display. We assumed that the antiviral directly inhibits the transition process of infected cells to infectious (virion producing) cells, i.e., acts on $\tau$. **Fig D. Impact of Medium Refresh.** Estimated inhibition percentage (A,B), percentage reduction in the basic reproduction number $R_0$ (C,D) and effective reproduction number $R_e$ (E,F) as a function of concentration, for the DENV-2/RL strain (A, C, E) and DENV-2/16681 strain (B, D, F) where the well medium was refreshed or not on day 4. Estimates were calculated by substituting 1,000 parameter values sampled from the posterior distribution into Eqs (1), (6) and (7) in the main text. Solid lines represent the median, the shading represents the 95% CrI, dotted grey vertical lines indicate the concentrations tested in the in vitro experiments, and the dotted blue and red vertical lines indicate the median concentration such the $R_e = 1$. The dotted black horizontal lines indicate the threshold for a 50% reduction (A,B,C,D), and when $R_e = 1$ (E,F). Here, measurements below the limit of quantification (crosses in Fig 1) were included during model fitting and we assume the antiviral directly inhibits the transition process of infected cells to infectious (virion producing) cells, i.e., acts on $\tau$. **Fig E. Cell Dynamics (DENV-2/16681 strain, no refresh).** Underlying modelled cell dynamics for the DENV-2/16681 strain with no medium refresh and antiviral concentrations of 0 nM, $2.56 \times 10^{-03}$ nM, $1.28 \times 10^{-02}$ nM, $6.40 \times 10^{-02}$ nM, 0.32 nM, 1.6 nM, and 8 nM. The vertical red line indicates the time the viral inoculum was added to each well. The modelled dynamics of the target cells are in blue, those of the infected cells are in grey and those of the infectious (virion producing) cells are in orange; solid lines represent the median and the shading represents the 95% CrI. We assumed that the antiviral directly inhibits the transition process of infected cells to infectious (virion producing) cells, i.e., acts on $\tau$. **Fig F. Model Fits (DENV-2/RL strain, no refresh).** Underlying modelled cell dynamics for the DENV-2/RL strain with no medium refresh and antiviral concentrations of 0 nM, $2.56 \times 10^{-03}$ nM, $1.28 \times 10^{-02}$ nM, $6.40 \times 10^{-02}$ nM 0.32 nM, 1.6 nM, and 8 nM. The vertical red line indicates the time the viral inoculum was added to each well. The modelled dynamics of the target cells are in blue, those of the infected cells are in grey and those of the infectious (virion producing) cells are in orange; solid lines represent the median and the shading represents the 95% CrI. We assumed that the antiviral directly inhibits the transition process of infected cells to infectious (virion producing) cells, i.e., acts on $\tau$. **Fig G. Goodness of Fit.** Observed vs predicted extracellular and intracellular RNA values for the experimental infection studies conducted,

disaggregated by JNJ-1802 concentration and DENV-2 strain. Here, intracellular measurements below the limit of quantification (LOQ) were included during model fitting and measurements below LOD were left-censored at the LOD during model fitting. We assumed that the antiviral directly inhibits the transition process of infected cells to infectious (virion producing) cells i.e., acts on $\tau$. For the observed values, we plot the median observed value across individual wells and the corresponding 2.5–97.5 percentiles. For the predicted values, we plot the median predicted value and corresponding 95% credible interval. For both observed and predicted values, we plot values below the LOD at the LOD (dashed red horizontal and vertical lines) to aid visual comparison between the observed and predicted values. **Fig H. Model Fits (DENV-2/16681 strain, no refresh).** Model fits for the measurements observed using the DENV-2/16681 strain with no medium refresh and antiviral concentrations of 0 nM, $2.56\times10^{-03}$ nM, $1.28\times10^{-02}$ nM, $6.40\times10^{-02}$ nM, 0.32 nM, 1.6 nM, and 8 nM. Coloured points represent the data from each well (6 wells for 0 nM, 4 wells for concentrations >0 nM), the vertical red line indicates the time the viral inoculum was added to each well and the horizontal dashed black line indicates the limit of quantification. The modelled dynamics of the intracellular RNA virus are in green and those of the extracellular RNA virus are in purple; solid lines represent the median and the shading represents the 95% CrI. Viral suppression is observed for concentrations of 1.6 nM and 8 nM. Here, measurements were left-censored at the LOQ during model fitting and we plot measurements below the LOQ (crosses in Fig 1) at the LOQ for visual display. We assumed that the antiviral inhibits the transition process of infected cells to infectious (virion producing) cells, i.e. acts on $\tau$. **Fig I. Model Fits (DENV-2/RL strain, no refresh).** Model fits for the measurements observed using the DENV-2/RL strain with no medium refresh and antiviral concentrations of 0 nM, $2.56\times10^{-03}$ nM, $1.28\times10^{-02}$ nM, $6.40\times10^{-02}$ nM, 0.32 nM, 1.6 nM, and 8 nM. Coloured points represent the data from each well (6 wells for 0 nM, 4 wells for concentrations >0 nM), the vertical red line indicates the time the viral inoculum was added to each well and the horizontal dashed black line indicates the limit of quantification. The modelled dynamics of the intracellular RNA virus are in green and those of the extracellular RNA virus are in purple; solid lines represent the median and the shading represents the 95% CrI. Viral suppression is observed for concentrations of 1.6 nM and 8 nM. Here, measurements were left-censored at the LOQ during model fitting and we plot measurements below the LOQ (crosses in Fig 1) at the LOQ for visual display. We assumed that the antiviral inhibits the transition process of infected cells to infectious (virion producing) cells, i.e. acts on $\tau$. **Fig J. Limit of Quantification (No Refresh).** Estimated inhibition percentage (A,B), percentage reduction in the basic reproduction number $R_0$ (C,D) and effective reproduction number $R_e$ (E,F) as a function of concentration for the DENV-2/RL strain (A,C,E) and DENV-2/16681 strain (B,D,F) where measurements were left-censored at either the limit of detection (LOD) or the limit of quantification (LOQ) and the well medium was not refreshed. Estimates were calculated by substituting 1,000 parameter values sampled from the posterior distribution into Eqs (1), (6) and (7) in the main text. Solid lines represent the median, the shading represents the 95% CrI, dotted grey vertical lines indicate the concentrations tested in the in vitro experiments, and the dotted blue and red vertical lines indicate the median concentration such the $R_e$ = 1. The dotted black horizontal lines indicate the threshold for a 50% reduction (A,B,C,D), and when $R_e$ = 1 (E,F). Here, we assume the antiviral directly inhibits the transition process of infected cells to infectious (virion producing) cells, i.e. acts on $\tau$. **Fig K. Model Schematic.** Target cells are infected at a rate $\beta$ per virion. Following a latent period of $1/\tau$ days on average, infectious cells $I$ synthesize intracellular viral RNA ($X$) at a net production rate $\omega$ (in absence of antiviral drug), a proportion p of which then gets secreted as extracellular viral RNA at a rate $\varepsilon$. Extracellular viral RNA decays at a rate $\kappa_V$. Target cells replicate at a rate $s_N$; target cells and infected cells have a mean lifespan of $1/\kappa_T$ and $1/(\kappa_T + \kappa_I)$

days, respectively. Individual wells have a carrying capacity $K_W$. The antiviral molecule acts on intracellular RNA production, following a concentration-dependent function f(C). The parameters in red are estimated, those in black are fixed. **Fig L. Cell Dynamics (DENV-2/16681 strain, no refresh).** Underlying modelled cell dynamics for the DENV-2/16681 strain with no medium refresh and antiviral concentrations of 0 nM, $2.56 \times 10^{-03}$ nM, $1.28 \times 10^{-02}$ nM, $6.40 \times 10^{-02}$ nM, 0.32 nM, 1.6 nM, and 8 nM. The vertical red line indicates the time the viral inoculum was added to each well. The modelled dynamics of the target cells are in blue, those of the infected cells are in grey and those of the infectious (virion producing) cells are in orange; solid lines represent the median and the shading represents the 95% CrI. We assumed that the antiviral directly inhibits intracellular RNA production, i.e., acts on ω. **Fig M. Cell Dynamics (DENV-2/RL strain, no refresh).** Underlying modelled cell dynamics for the DENV-2/16681 strain with no medium refresh and antiviral concentrations of 0 nM, $2.56 \times 10^{-03}$ nM, $1.28 \times 10^{-02}$ nM, $6.40 \times 10^{-02}$ nM, 0.32 nM, 1.6 nM, and 8 nM. The vertical red line indicates the time the viral inoculum was added to each well. The modelled dynamics of the target cells are in blue, those of the infected cells are in grey and those of the infectious (virion producing) cells are in orange; solid lines represent the median and the shading represents the 95% CrI. We assumed that the antiviral directly inhibits intracellular RNA production, i.e., acts on ω. **Fig N. Model Fits (DENV-2/16681 strain, no refresh).** Model fits for the measurements observed using the DENV-2/16681 strain with no medium refresh and antiviral concentrations of 0 nM, $2.56 \times 10^{-03}$ nM, $1.28 \times 10^{-02}$ nM, $6.40 \times 10^{-02}$ nM 0.32 nM, 1.6 nM, and 8 nM. Coloured points represent the data from each well (6 wells for 0 nM, 4 wells for concentrations >0 nM), the vertical red line indicates the time the viral inoculum was added to each well and the horizontal dashed black line indicates the limit of detection. The modelled dynamics of the intracellular RNA virus are in green and those of the extracellular RNA virus are in purple; solid lines represent the median and the shading represents the 95% CrI. Here, measurements below the limit of quantification (crosses in Fig 1) were included during model fitting. Measurements below the LOD were left censored at the LOD during model fitting and we plot these measurements at the LOD for visual display. We assumed that the antiviral directly inhibits intracellular RNA production, i.e., acts on ω. **Fig O. Model Fits (DENV-2/RL strain, no refresh).** Model fits for the measurements observed using the DENV-2/RL strain with no medium refresh and antiviral concentrations of 0 nM, 2.56x10-03 nM, 1.28x10-02 nM, 6.40x10-02 nM, 0.32 nM, 1.6 nM, and 8 nM. Coloured points represent the data from each well (6 wells for 0 nM, 4 wells for concentrations >0 nM), the vertical red line indicates the time the viral inoculum was added to each well and the horizontal dashed black line indicates the limit of detection. The modelled dynamics of the intracellular RNA virus are in green and those of the extracellular RNA virus are in purple; solid lines represent the median and the shading represents the 95% CrI. Here, measurements below the limit of quantification (crosses in Fig 1) were included during model fitting. Measurements below the LOD were left censored at the LOD during model fitting and we plot these measurements at the LOD for visual display. We assumed that the antiviral directly inhibits intracellular RNA production, i.e., acts on ω. **Fig P. Effect of antiviral (No Refresh).** Estimated inhibition percentage (A,B), percentage reduction in the basic reproduction number R0 (C,D) and effective reproduction number Re (E,F) as a function of concentration, for the DENV-2/RL strain (A,C,E) and DENV-2/16681 strain (B,D,F) where the drug action was on intracellular RNA production in a model with/without a latent period (WL/NL) allowing for maturation of infected cells (action on ω) or the drug action was on the transition process of infected cells to infectious (virion producing) cells (action on τ) and the well medium was not refreshed. Estimates were calculated by substituting 1,000 parameter values sampled from the posterior distribution into Eqs (1), (6) and (7) in the main text and above. Solid lines represent the median, the shading represents the 95% CrI,

dotted grey vertical lines indicate the concentrations tested in the in vitro experiments, and the dotted green, blue and red vertical lines indicate the median concentration such the Re = 1. The dotted black horizontal lines indicate the threshold for a 50% reduction (A,B,C,D), and when Re = 1 (E,F). Here, measurements below the limit of quantification (crosses in Fig 1) were included during model fitting. **Table A. Posterior parameter estimates.** Median posterior estimate and 95% credible interval (CrI) in brackets. Here the measurements below the limit of quantification (crosses in Fig 1) were included during model fitting, and we assumed that the antiviral directly inhibits transition of infected cells to infectious (virion producing) cells, i.e., acts on τ. **Table B. Posterior Parameter Estimates.** Posterior parameter estimates for models assuming left-censoring of the viral RNA data at the limit of detection or at the limit of quantification. The median posterior estimate is reported with the 95% credible interval (CrI) in brackets. Here, we assume the antiviral directly inhibits the transition process of infected cells to infectious (virion producing) cells, i.e., acts on τ. **Table C. Posterior Parameter Estimates.** Posterior parameter estimates for different modes of drug action and model structure–(1) inhibition of transition process of infected cells to infectious (virion producing) cells (action on τ), (2) direct inhibition of intracellular RNA production in a model with no latent period allowing for the transition process of infected cells to infectious cells (action on ω, no latent period) and (3) direct inhibition of intracellular RNA production in a model with a latent period allowing for the transition process of infected cells to infectious cells (action on ω, with latent period) The median posterior estimate is reported with the 95% credible interval in brackets. Here, measurements below the limit of quantification but above the limit of detection (crosses in Fig 1) were included during model fitting. **Table D. Posterior Parameter Estimates (RL Strain).** Posterior parameter estimates for different starting values of the initial viral inoculum (V0). Here, measurements below the limit of quantification (crosses in Fig 1) were included during model fitting. Measurements below the LOD were left censored at the LOD during model fitting and we assumed that the antiviral directly inhibits the transition process of infected cells to infectious (virion producing) cells, i.e., acts on τ. **Table E. Posterior Parameter Estimates (16681 Strain).** Posterior parameter estimates for different starting values of the initial viral inoculum (V0). Here, measurements below the limit of quantification (crosses in Fig 1) were included during model fitting. Measurements below the LOD were left censored at the LOD during model fitting and we assumed that the antiviral directly inhibits the transition process of infected cells to infectious (virion producing) cells, i.e., acts on τ. **Table F. Posterior Parameter Estimates (RL Strain).** Sensitivity of posterior parameter estimates to data excluded during model fitting. Here, measurements below the limit of quantification (crosses in Fig 1) were included during model fitting. Measurements below the LOD were left censored at the LOD during model fitting and we assumed that the antiviral directly inhibits the transition process of infected cells to infectious (virion producing) cells, i.e., acts on τ. **Table G. Posterior Parameter Estimates (16681 Strain).** Sensitivity of posterior parameter estimates to data excluded during model fitting. Here, measurements below the limit of quantification (crosses in Fig 1) were included during model fitting. Measurements below the LOD were left censored at the LOD during model fitting and we assumed that the antiviral directly inhibits the transition process of infected cells to infectious (virion producing) cells, i.e., acts on τ.
(PDF)

## Acknowledgments

This work benefitted from discussions held with Liesbeth Van Wesenbeeck and Marjolein Crabbe (Janssen Research & Development).

## Author Contributions

**Conceptualization:** Neil M. Ferguson, Oliver Ackaert, Ilaria Dorigatti.

**Data curation:** Olivia Goethals, Peggy Geluykens, Doortje Borrenberghs, Oliver Ackaert.

**Formal analysis:** Clare P. McCormack.

**Investigation:** Olivia Goethals, Peggy Geluykens, Doortje Borrenberghs, Oliver Ackaert.

**Methodology:** Clare P. McCormack, Olivia Goethals, Nele Goeyvaerts, Xavier D. Woot de Trixhe, Neil M. Ferguson, Oliver Ackaert, Ilaria Dorigatti.

**Software:** Clare P. McCormack.

**Validation:** Clare P. McCormack, Olivia Goethals, Nele Goeyvaerts, Xavier D. Woot de Trixhe, Neil M. Ferguson, Oliver Ackaert, Ilaria Dorigatti.

**Visualization:** Clare P. McCormack.

**Writing – original draft:** Clare P. McCormack, Ilaria Dorigatti.

**Writing – review & editing:** Clare P. McCormack, Olivia Goethals, Nele Goeyvaerts, Xavier D. Woot de Trixhe, Peggy Geluykens, Doortje Borrenberghs, Neil M. Ferguson, Oliver Ackaert, Ilaria Dorigatti.

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
