## [Decision Letter · Decision Letter 0]

31 Aug 2023

Dear Dr McCormack,

Thank you very much for submitting your manuscript "Modelling the impact of JNJ-1802, a first-in-class dengue inhibitor blocking the NS3-NS4B interaction, on in-vitro DENV-2 dynamics" for consideration at PLOS Computational Biology. As with all papers reviewed by the journal, your manuscript was reviewed by members of the editorial board and by several independent reviewers. The reviewers appreciated the attention to an important topic. Based on the reviews, we are likely to accept this manuscript for publication, providing that you modify the manuscript according to the review recommendations.

Sincerely,

Alex Perkins

Academic Editor

PLOS Computational Biology

Rob De Boer

Section Editor

PLOS Computational Biology

Reviewer's Responses to Questions

**Comments to the Authors:**

Reviewer #1: This is a nicely done analysis on an interesting topic. The paper is well written and clear. I have just a couple of minor comments/suggestions.

Introduction:

Line 72: extra “I” typo

With the results before methods order it would be helpful to have a brief description of the model at the start of the results

Suggest replacing e with 10 in the numbers..

Line 257: “amounting to” – found this wording a little confusing- suggest rephrasing

Discussion:

Suggest adding something on the timing of administration- even with excellent inhibition what can the impact be if given later in infection?

Reviewer #2: The review is uploaded as an attachment.

**Have the authors made all data and (if applicable) computational code underlying the findings in their manuscript fully available?**

Reviewer #1: Yes

Reviewer #2: Yes

PLOS authors have the option to publish the peer review history of their article (what does this mean?). If published, this will include your full peer review and any attached files.

Reviewer #1: No

Reviewer #2: No

Figure Files:

Data Requirements:

Reproducibility:

References:

---

## [Editor Report · Decision Letter 1]

5 Nov 2023

Dear Dr McCormack,

We are pleased to inform you that your manuscript 'Modelling the impact of JNJ-1802, a first-in-class dengue inhibitor blocking the NS3-NS4B interaction, on in-vitro DENV-2 dynamics' has been provisionally accepted for publication in PLOS Computational Biology.

Best regards,

Alex Perkins

Academic Editor

PLOS Computational Biology

Rob De Boer

Section Editor

PLOS Computational Biology

---

## [Editor Report · Acceptance letter]

17 Nov 2023

PCOMPBIOL-D-23-01167R1 

Modelling the impact of JNJ-1802, a first-in-class dengue inhibitor blocking the NS3-NS4B interaction, on in-vitro DENV-2 dynamics

Dear Dr McCormack,

I am pleased to inform you that your manuscript has been formally accepted for publication in PLOS Computational Biology. Your manuscript is now with our production department and you will be notified of the publication date in due course.

With kind regards,

Anita Estes
